# Consumption of protect and risk food groups for non-communicable diseases among women of reproductive age in rural areas of Morogoro, Tanzania

**Akwilina Wendelin Mwanri**[1]*, **Fulgence Mishili**[2], **Theresia Assenga**[2], **Rashid Suleiman**[3], **Dismas Mwaseba**[4], **Susan Nchimbi-Msolla**[5]

1 Department of Human Nutrition and Consumer Sciences, Sokoine University of Agriculture, Morogoro, Tanzania, 2 Department of Agricultural Economics and Agribusiness, Sokoine University of Agriculture, Morogoro, Tanzania, 3 Department of Food Science and Agro-processing, Sokoine University of Agriculture, Morogoro, Tanzania, 4 Department of Agricultural Extension and Community Development, Sokoine University of Agriculture, Morogoro, Tanzania, 5 Department of Crop Science and Horticulture, Sokoine University of Agriculture, Morogoro, Tanzania

☯ These authors contributed equally to this work.
* akwmwanri@sua.ac.tz

## Abstract

Malnutrition has been persistently high in Tanzania, especially in the rural areas. Although some interventions have been implemented to reduce the rate of undernutrition, improvement is much slower than expected. Moreover, the prevalence of overweight and obesity among women of reproductive age has been on the increase. Changes in lifestyles specifically in eating habits may contribute to the observed increase trend in the prevalence of overweight and obesity. However, there are limited studies that have assessed diet quality for women. Hence, this study aimed to assess the consumption of protective and risk food groups for non-communicable diseases (NCDs) in rural areas of the Morogoro region, Tanzania. The cross-sectional study involved 512 randomly selected women of reproductive age in Mvomero district. All mothers or caretakers with children of age 6–23 months who resided in the area for over three months were eligible to participate. A structured questionnaire was used to collect demographic data and the Global Diet Quality Questionnaire was used to collect dietary intake data for women. A multivariate regression model was used to identify factors associated with consumption of diversified diets, NCD-protect, and NCD-risk food group scores. The results show that about 42% of the women had no formal education and about one in three women own a mobile phone. About 70% consumed vegetables while 33% consumed deep fried foods. Only 34% of the women met the minimum diet diversity (MDD-W) of five or more food groups. The mean NCD-protect score was 2.8±1.4 and the NCD-risk score was 0.77±0.97. Household income was positively associated with an increase in both protect scores and NCD-risk. Mobile phone ownership was positively associated with NCD-risk. Other factors such as age and self-employment in agriculture were positively associated NCD-protect while the number of children was negatively associated with NCD-protect score. Rural communities should be educated on the importance of food diversification starting from production to

**Data availability statement:** The data can be accessed using the following link: (https://www.taccire.sua.ac.tz/handle/123/564).

**Funding:** We declare that the research was funded by European Union's Horizon 2020 research and innovation programme under grant agreement [GA No 862802]. However, the funders had no role in study design, data collection and analysis, decision to publish, or preparation of the manuscript.

**Competing interests:** The authors have declared that no competing interests exist.

consumption and the need for multiple sources of income to increase purchasing power of the household while considering proper food choices to avoid the risk of NCDs. The Ministry of Agriculture and other stakeholders should support and promote production of bio-fortified crops and keeping small animals. The Ministry of Education should ensure that school children are aware of the NCD-risk foods and make available healthier food choices around the school environment.

## Introduction

Despite the great progress made in improving food and nutrition security and reducing poverty over the past decades, undernutrition is still affecting billions of people worldwide [1]. Globally, about 170 million women are underweight; while three times as many women (610 million) are overweight [2]. Each year, about 20 million babies are born with low birth weight, which is closely associated with the eating behaviors and nutrition status of women before and during pregnancy [3]. Although undernutrition affects both urban and rural low-income households, those in the rural areas face more challenges than their urban counterparts. This is due to social isolation, intermittent drought, limited participation of women in economic activities, limited market access, inadequate access to health care, and other socioeconomic factors such as a lower education level and a high unemployment rate [4].

In Tanzania, undernutrition affects infants and children under the age of five, as well as women of childbearing age [5]. Currently, about 7.3% of non-pregnant women aged 15–49 years are underweight [6]. According to the Tanzania Demographic Health Survey and Malaria Indicator Survey, the prevalence of chronic malnutrition [stunting] among children below five years in the Morogoro region is high, at about 31% which is similar to the national average [6]. The prevalence of acute malnutrition is 4% which is above the national average and underweight is 10% [6] Studies show that undernutrition is particularly high among low-income rural Tanzanian households, mainly because they consume more carbohydrate-rich, staple-based diets that are low in minerals and vitamins [4,7]. Other factors contributing to malnutrition include frequent illness, low birth weight and inadequate childcare practices. In addition, only 46% of rural women in Tanzania meet their minimum dietary diversity requirement (i.e., consume at least 5 food groups out of 10 daily), and women's intake of protein, fruits, and vegetables is very low [4].

Rapid urbanization, along with the global economic crisis in developing countries, has put many urban and rural people at high risk of bad quality diets and malnutrition. There is a shift from consumption of traditional diets to more refined foods high in fat, sugar and salt even in the rural areas. In fact, many studies have shown that the population of women of reproductive age is at high risk of specific micronutrient deficiencies, which can affect their health [8–11]. Generally, the diets of Tanzanian women are less diversified and have a low intake of protein, fruits and vegetables [4]. Another study by Paul et al (2022) in Tanzania found that dietary quality was inversely associated with the risk of overweight and obesity [12]. Although there is an increased prevalence of overweight or obesity among women of reproductive age from 11% in 1991 to 36% in 2022 [6], most of the reported studies concentrated on assessing nutritional status and dietary diversity with limited information on unhealthy diets. Hence this study sought to assess the consumption of foods that increase the risk and those that are protective against diet-related NCDs among rural women of reproductive age.

## Materials and methods

### Description of the study area

A cross-sectional study was conducted in Mvomero District, Morogoro, specifically in the rural setting. Mvomero is one of the seven districts in Morogoro Region. The district is characterized by an average annual rainfall ranging between 487 and 1951 mm per year [13]. Based on the current population census, the total population for the district is 421,741, with 210,834 males and 210,907 females [14]. The district lies in three ecological zones: Miombo woodland, Highland, and mountains, as well as the Savanna River basin.

Some of the economic activities taking place in the area include agriculture, which employs about 75% of the district residents. Majority are small-scale farmers who produce various crops including maize, millet, cassava, sweet potatoes, beans, rice, sugar cane, cotton, and vegetables. Other activities done with the percentage of those involved in brackets are mining and quarrying employing (5%), trade and commerce (5%), and livestock keeping, fishing and hunting (3%). During the dry season, the majority of agricultural activities depend on the irrigation schemes established in the district which include Mkindo, Mlali, Mgeta, Ndole, Dihombo, Dakawa, and Komtonga irrigation schemes [13].

### Data collection and sampling

Data were collected from November to December, 2022 from six villages located in Kinda Ward, namely Kinda, Magunga, Makuyu, Matale, Ndole, and Semwali. The sampling involved 518 randomly selected mothers with children of age between 6 and 23 months. The mothers were selected randomly from the village registry which had a list of all women with children of age 6 to 23 months. Six mothers/caretakers did not complete the study because they missed the dietary intake data. All women who resided in the area for over three months were eligible to participate. Women known to be pregnant were excluded from the study because there could be pregnancy induced changes in food consumption. A structured, pretested questionnaire translated from English to Kiswahili was administered to collect data on socio-demographic characteristics, food availability, monthly income expenditure on food and decision-making in the household. In addition, the Global Diet Quality Questionnaire (GDQQ)—adapted for Tanzania for women was used to determine food group consumption, a medium dietary diversity score for women (MDD-W) and NCD-protect and NCD-risk scores based on 24-hr dietary recall [15]. The tools are available online, translated into Kiswahili and are free to be used by researchers [15]. The enumerator read the examples of the foods for each food group and marked one if any of the foods on the list within the food group were consumed or zero if none were consumed.

The 29 food groups from the GDQQ were aggregated to form nine NCD-protect and eight NCD-risk food groups. The NCD-Protect score (0-9) was based on food consumption from nine healthy food groups, namely whole grains, pulses, nuts and seeds, vitamin A-rich orange vegetables, dark green leafy vegetables, other vegetables, vitamin A-rich fruits, citrus, and other fruits [15]. A higher score indicates the inclusion of more health-promoting foods in the diet and correlates positively with meeting global dietary recommendations. Similarly, the NCD-risk score is a score with a range from 0 to 9. The score was based on the consumption of foods from eight food groups that are recommended to limit or avoid during the past day and night (one food group, processed meat, is double-weighted). The other food groups are soft drinks (sodas), baked / grain-based sweets, other sweets, unprocessed red meat, deep-fried food, fast food, instant noodles, and packaged ultra-processed salty snacks. A higher score indicates higher consumption of foods and drinks to avoid or limit.

## Data analysis

STATA Statistical software was employed to analyze demographic factors, women's dietary diversity, and women's NCD- protect and risk scores, and the results were reported as descriptive. Some of the demographic factors used include monthly household income, a share of income spent on food, food shortage, mobile ownership, self-employment in agriculture, age, education level, sex of the household head, and respondents' number of children. Categorical variables were dichotomized and coded as one for those with the characteristics and zero for those without while discrete variables such as household income and number of children were used as continuous variables.

Econometric model The study used a multivariate regression model to assess the determinants of women's NCD-protect and risk scores. The model is an extension of a simple linear regression model with the addition of more than one explanatory variable and outcome variables [16]. It can be used when the dependent variables are affected or influence by common independent variables. During analysis, two models with two dependent variables, that is, NCD-protect or risk scores, were analyzed with independent variables as indicated in equations 1 and 2. This study assumed that NCD-protect and risk scores can be influenced by common factors such as household monthly income, a share of income spent on food, food shortage, mobile ownership, employment, self-employed in agriculture (agricultural), age, education level, sex of the household head, and respondents' number of children. The variable of household income was transformed into a log to ensure data are normally distributed.

NCD Protect cores

$$
\begin{aligned}
=&\beta_0 + B_1 \text{ Monthly Household Income} + \beta_2 \text{Mother's age is above 35 Years} \\
&+ \beta_3 \text{Mother's age is between 26 to 35 Years} \\
&+ \beta_4 \text{ Share of food expenditure is about 50\%} \\
&+ \beta_5 \text{Share food expenditure is above 50\%} + \beta_6 \text{ Total number of Children} \\
&+ \beta_7 \text{ Secondary Education and Above} + \beta_8 \text{Primary Education} \\
&+ \beta_9 \text{ Agricultural} + \beta_{10} \text{Mobile Access} + \beta_{11} \text{Sex of Household Head is male} \\
&+ \varepsilon \ldots\ldots\ldots\ldots
\end{aligned}
\tag{1}
$$

$$
\begin{aligned}
\text{NCD Risk} =&\beta_0 + B_1 \text{ Monthly Household Income} + \beta_2 \text{Mother's age is above 35 Years} \\
&+ \beta_3 \text{Mother's age is between 26 to 35 Years} \\
&+ \beta_4 \text{ Share of food expenditure is about 50\%} \\
&+ \beta_5 \text{Share of food expenditure is above 50\%} + \beta_6 \text{ Total number of Children} \\
&+ \beta_7 \text{ Secondary Education and Above} + \beta_8 \text{Primary Education} + \beta_9 S \\
&+ \beta_{10} \text{Mobile Access} + \beta_{11} \text{Sex of Household Head} \\
&+ \varepsilon \ldots\ldots\ldots\ldots
\end{aligned}
\tag{2}
$$

## Minimum dietary diversity for women

Ten food groups that include, "grains (white roots and tubers, and plantains), pulses (beans, peas and lentils), nuts and seeds, dairy, meat (poultry and fish), eggs, dark green leafy vegetables, other vitamin A-rich fruits and vegetables, other vegetables and fruits" were used to determine the MDD-W [17,18]. Each food was assigned a weight of 1 if consumed and 0 otherwise to give a total of 10 if all food groups were consumed in the 24 hours preceding the survey [15,17,18]. Women who consumed five or more food groups were regarded as attaining a minimum dietary diversity (1), while those who consumed less than five food groups were

considered otherwise (0). A binary logit regression model with a dependent variable of women's minimum dietary diversity was used to assess the influence of social-demographic factors on women's dietary diversity.

## Ethical consideration

Permission to conduct the study was sought from Sokoine University of Agriculture and the regional, district, ward, and village authorities. The study was approved and ethical clearance obtained from National Health Research Ethics Committee of the Tanzania National Institute for Medical Research (NIMR) with reference number NIMR/HQ/R.8a/VOL.IX/3605. Participants were informed about the objectives of the study, the procedure, benefits, and risks. Written consent was obtained from those who agreed to participate. Participation was voluntary and each participant was free to decide otherwise at any time. All participants were identified by code numbers; data was kept in a secured computer and were accessible to the research team only.

## Results

The mean age of the respondents was 26.4 (SD7.3), ranging from 15 to 49 years. More than 50% of the respondents were of age below 25 years. About 42% of the respondents did not have any formal education, and one in three owned a mobile phone. About 45% spent more than half of their income on food (Table 1).

## Descriptive results for numeric variables

The mean monthly household income was 112,569TZS (47USD) ranging from 3000TZS (1.3USD) to 1,550,000 TZS (633USD). The average number of children per household was about three children, ranging from one to eleven children (Table 2).

**Table 1. Summary of descriptive analysis for demographic and other factors (N = 512).**

| Variables | | n (%) |
|---|---|---|
| Age class | 15–25 years | 284 (55) |
| | 26–35years | 162 (32) |
| | Above 35 years | 66 (13) |
| Share of income on food | Less than 50 | 142 (28) |
| | 50% Share | 141 (28) |
| | Above 50 | 229 (45) |
| Mobile ownership | No | 337 (66) |
| | Yes | 175 (34) |
| Education level | No formal education | 215 (42) |
| | Primary education | 265 (52) |
| | Secondary and above | 32 (6) |
| Agricultural | No | 47 (9) |
| | Yes | 465 (91) |
| Gender of the household head | Male household head | 464 (91) |
| | Female household head | 48 (9) |

**Table 2. Descriptive results for numeric variables.**

| Variable | Observation | Mean | Std. Dev. | Min | Max |
|---|---|---|---|---|---|
| Household income (TZS) | 512 | 112,569 (47USD) | 138,835 (57USD) | 3000TZ (1.3USD) | 1,550,000TZS (633USD) |
| Number of children | 512 | 3 | 2 | 1 | 11 |

## Consumption of NCD-protect and NCD-risk food groups (N = 512)

Table 3 shows consumption of food groups that are considered healthy or unhealthy for women. Regarding healthy foods, vegetables and pulses were commonly consumed by a large proportion of women (72% and 44%), respectively. However, only 16% of women consumed whole grains. Among the unhealthy foods, relatively high proportion of women consumed deep fried foods (33%), and baked/grain based sweet foods (15%) were consumed as unhealthy food.

## Descriptive analysis for women NCD protect and risk scores

The mean NCD-protect score was $2.8 \pm 1.4$, with a range from 0 to 8 food groups while the NCD-risk score around $0.7 \pm 0.9$ with a range of 0 to 5 food groups (Table 4).

The mean NCD-protect score was higher among women who spent about half of their income on food (mean $3.0 \pm 1.5$) and higher among women with a single child ($0.9 \pm 1.0$) (Table 4).

## Influence of demographic factors to women NCD- protect and NCD-risk scores

The multivariate regression model shows that there was a significant association between the NCD-protect score and household income, age (above 35 years), number of children in the household, agricultural employment, and sex of the household head. There was a significant positive association between household income and mobile phone ownership with NCD-risk score (Table 5).

## Minimum dietary diversity for women and its determinants

Tables 6 and 7 show that only one-third of the women (34%, n = 174) attained the MDD-W. Household income, age of the mother, number of children, ownership of a mobile phone and sex of the household head were associated with the MDD-W

**Table 3. Consumption of NCD-protect and NCD-risk food groups (N = 512).**

| NCD-Protect food groups | n (%) | NCD Risk food groups | n (%) |
|---|---|---|---|
| Whole grains | 82 (16) | Sugar sweetened beverages | 28 (6) |
| Pulse | 226 (44) | Baked/grains based sweets | 77 (15) |
| Nuts and seeds | 54 (11) | Other sweet | 45 (9) |
| Vitamin A rich orange vegetables | 16 (3) | Processed meat | 0 (0) |
| Dark green leafy vegetables | 361 (71) | Unprocessed red meat | 38 (7) |
| Other vegetables | 370 (72) | Deep-fried foods | 168 (33) |
| Vitamin A rich fruits | 126 (25) | Fast foods and instant noodles | 2 (0.03) |
| Citrus | 17 (3) | Processed packed salty snacks | 2 (0.03) |
| Other fruits | 164 (32) | | |

**Table 4.** Descriptive analysis for women NCD-protect (mean+/−SD) and NCD-risk (mean+/−SD) Scores.

| Variables | | Observations | *NCD-Protect | *NCD-Risk |
|---|---|---|---|---|
| Age class | 15–25 years | 282 | 2.8 ± 1.4 | 0.8 ± 0.96 |
| | 26–35 years | 163 | 2.7 ± 1.5 | 0.7 ± 1.1 |
| | Above 35 years | 67 | 2.6 ± 1.4 | 0.5 ± 0.8 |
| Number of children | 1 child | 161 | 3.0 ± 1.5 | 0.9 ± 1.0 |
| | 2 to 3 children | 208 | 2.7 ± 1.4 | 0.7 ± 1.0 |
| | Above 3 children | 143 | 2.6 ± 1.4 | 0.6 ± 1.0 |
| Share of income on food | Less than 50% | 142 | 2.9 ± 1.6 | 0.8 ± 1.0 |
| | About 50% | 141 | 3.0 ± 1.5 | 0.76 ± 0.9 |
| | Above 50% | 229 | 2.6 ± 1.3 | 0.6 ± 0.98 |
| Mobile ownership | No | 337 | 2.7 ± 1.5 | 0.6 ± 0.9 |
| | Yes | 175 | 2.9 ± 1.4 | 0.9 ± 1.1 |
| Education level | No formal education | 215 | 2.7 ± 1.4 | 0.7 ± 1.0 |
| | Primary education | 266 | 2.8 ± 1.5 | 0.7 ± 1.0 |
| | Secondary and above | 31 | 2.7 ± 1.3 | 0.9 ± 0.9 |
| agricultural | No | 47 | 2.4 ± 1.3 | 0.7 ± 0.8 |
| | Yes | 465 | 2.8 ± 1.5 | 0.7 ± 1.0 |
| Gender of the household head | Male household head | 463 | 2.8 ± 1.4 | 0.7 ± 1.0 |
| | Female household head | 49 | 2.4 ± 1.4 | 0.7 ± 0.8 |
| Total | | | 2.8 ± 1.4 | 0.71 ± 0.97 |

*Value in the table mean ± standard deviation.

**Table 5.** Influence of demographic factors to women NCD- Protect and risk scores.

| Variables | NCD Protect | | | NCD Risk | | |
|---|---|---|---|---|---|---|
| | Coefficients | S.E | P>t | Coefficients | S.E | P>t |
| Household income (Log) | 0.299 | 0.059 | 0.000*** | 0.254 | 0.040 | 0.000*** |
| Above 35 years | 0.784 | 0.316 | 0.013** | −0.043 | 0.213 | 0.841 |
| Age 26 to35 years | 0.293 | 0.184 | 0.112 | 0.012 | 0.124 | 0.923 |
| About 50% share of income on food | 0.168 | 0.171 | 0.326 | 0.052 | 0.115 | 0.65 |
| Above 50% share of income on food | −0.248 | 0.152 | 0.103 | −0.142 | 0.102 | 0.165 |
| Number of children | −0.205 | 0.057 | 0.000*** | −0.041 | 0.039 | 0.285 |
| Secondary education and above | −0.198 | 0.297 | 0.506 | 0.043 | 0.200 | 0.83 |
| Primary Education | 0.063 | 0.136 | 0.646 | −0.142 | 0.092 | 0.123 |
| Agricultural | 0.458 | 0.226 | 0.043** | 0.108 | 0.152 | 0.48 |
| Mobile Ownership | 0.084 | 0.141 | 0.55 | 0.211 | 0.095 | 0.027** |
| Sex of Household head | 0.369 | 0.222 | 0.097* | 0.016 | 0.150 | 0.916 |
| Constant | −0.901 | 0.724 | 0.214 | −2.044 | 0.488 | 0.000** |
| Equation | RMSE | R-sq. | F | | | |
| NCD Protect | 1.40 | 0.10 | 4.90** | | | |
| NCD Risk | 0.94 | 0.11 | 5.35** | | | |

*** Significant at 1%, ** Significant at 5% and * Significant at 10%.

**Table 6. Minimum dietary diversity for women.**

| Women dietary diversity | N | % | Mean | S.D | Min | Max |
|---|---|---|---|---|---|---|
| MDD-W < 5 | 338 | 66 | 3 | 1 | 1 | 4 |
| MDD-W ≥ 5 | 174 | 34 | 6 | 1 | 5 | 10 |

**Table 7. Determinants of women minimum dietary diversity.**

| Independent Variables | Odds Ratio | Robust S.E | P > z | 95% Conf. Interval | |
|---|---|---|---|---|---|
| Household income (log) | 1.6847 | 0.1615 | 0.000*** | 1.3961 | 2.0330 |
| Above 35 years | 3.8429 | 2.0441 | 0.011** | 1.3549 | 10.8999 |
| Age 26 to 35 years | 1.7504 | 0.5243 | 0.062 | 0.9732 | 3.1485 |
| About 50% share of income on food share | 1.0886 | 0.2956 | 0.755 | 0.6393 | 1.8535 |
| Above 50% share of income on food | 0.6727 | 0.1643 | 0.105 | 0.4168 | 1.0858 |
| Number of children | 0.7084 | 0.0710 | 0.001*** | 0.5820 | 0.8623 |
| Secondary education and Above | 0.9968 | 0.4552 | 0.994 | 0.4073 | 2.4395 |
| Primary Education | 1.1997 | 0.2652 | 0.41 | 0.7779 | 1.8503 |
| Agricultural | 1.8369 | 0.6798 | 0.1 | 0.8894 | 3.7939 |
| Mobile Ownership | 1.7683 | 0.3981 | 0.011** | 1.1374 | 2.7490 |
| Sex of household head | 2.6168 | 1.0126 | 0.013** | 1.2257 | 5.5866 |
| Constant | 0.0006** | 0.0007** | 0.000** | 0.0001 | 0.0057 |

*** Significant at 1%, ** Significant at 5% and * Significant at 10%.

## Discussion

This study aimed to assess the consumption of food groups that increase the risk and those that are protective against diet related NCDs among women of reproductive age in rural areas of Morogoro, Tanzania. There are limited number of studies that used GDQQ to estimate minimum dietary diversity, NCD-protect and NCD-risk scores among women of reproductive age in Tanzania [based on 24-hr dietary recall] [15]. The findings of this study indicate that the mean NCD-protect and NCD-risk scores for women were low, but the mean NCD-protect score was higher than that of the risk score, implying that rural women are more protected from NCDs as far as food intake is concerned. The lower consumption of NCD-risk food groups observed in this study can possibly be due to the fact that women living in rural areas have limited access and low purchasing power and hence depend mainly on what they produce. This finding is consistent with a review done by Akumu et al [19], which shows that, the diet in Uganda is mainly composed of starch staples, especially roots, tubers, bananas, and legumes which are healthy foods. A study conducted in Peru among women of reproductive age reported a higher prevalence of consumption of unhealthy foods, where by 97% consumed sugar-sweetened beverages [20].

A high proportion of women consumed vegetables and pulses in the healthy food groups. The observed practice is not surprising because the villages involved in the study produce pulses and plenty of vegetables in the river valleys. This finding is in line with the one reported in a study conducted in rural areas of Kenya, which revealed that vegetables and pulses were mostly consumed by women in rural areas [21]. However, contrasting results have been shown in other studies with a shortfall of fruits and vegetable intake in the diet of women of reproductive age [22–24]. Nonetheless, the findings of this study are inconsistent with several other studies that reported that the most commonly consumed food groups by women of

reproductive age in rural areas were staple foods such as grains or cereal, roots and tubers [23,25,26]. These results can possibly be explained by seasonality, socioeconomic variation among the study groups, as well as the availability of these food groups in households where vegetables are commonly produced and accessed from the market at a low price. However, the reported consumption was qualitative hence not necessarily imply that the studied women meet recommended amount of vegetable intake.

Deep-fried foods and baked/grain-based sweets were the most commonly consumed unhealthy foods. The high consumption of deep-fried foods shows changes in food preparation methods from common boiling or steaming to deep frying due to the availability and access of cooking oil, and shifts in dietary preferences, taste/palatability and lifestyles. Other studies conducted in Tanzania reported similar findings that about 21% of the rural households consume ultra-processed foods and foods away from home which are high in fat, sugar and salt [27,28].

From regression analysis, household income was positively associated with both NCD-protect and NCD-risk scores. This means households with more income can access more diversified diets, with some being in the healthy and others in the unhealthy groups. Higher-income households often have more disposable income, making it easier for them to afford a wide variety of foods, including healthier and unhealthy options. Nevertheless, lower-income households often face greater challenges in accessing and affording healthier food options, which can lead to consumption of monotonous, less diversified foods based on what they produce or what they can afford. More food access and availability due to high income may not necessarily lead to healthier option. It was reported that nutritious foods eaten at home in Tanzania has not improved with rising of income [27]. A study done in Tanzania using Household Budget Survey data support the current findings that lower income households purchase fewer healthful foods compared with higher income households [29]. Another study conducted in the in Sub-Saharan Africa reported similar findings; lower food expenditure is likely to be a key contributor to less healthy food choices among lower socioeconomic groups [30].

In this study, being older than 35 years was associated with greater consumption of NCD-protect food groups than their younger counterparts. With age, people often make lifestyle changes to prioritize health and well-being, including dietary modifications, as they develop more established and health-conscious dietary habits, which can include a greater focus on consuming NCD-protect foods [31]. It could also be explained by their being more exposed to nutrition education compared to the younger women. Another reason might be health conditions, as older individuals may have pre-existing health conditions that necessitate dietary changes, encouraging the consumption of NCD-protect foods. The finding of the current study is in line with a study conducted in Australia and Nepal, which found that being older than 35 years had higher component scores in limiting consumption of NCD-risk food groups, such as limiting consumption of saturated fat and consuming low-fat milk [32,33].

Furthermore, there were a significant negative association between the NCD-protect score and the number of children in the households. This means households with fewer children had higher NCD-protect scores compared to those with more children. This implies that households with few children are more likely to diversify their diet. These findings are supported by results from a systematic review and meta-analysis conducted in Ethiopia, which reported that, pregnant women living in larger families were more likely to consume less diversified diets compared to those living in smaller families [34]. This can possibly be explained by the fact that families with more children may have to allocate their resources, including food, among a greater number of individuals, which could affect the quality and diversity of their diet and strain the economic resources of a household, limiting their ability

to purchase a variety of NCD-protect foods. It might as well be related to time constraints that make it more challenging to prepare and consume healthier meals, potentially leading to reliance on more convenient but less nutritious options. Despite this, smaller families might have more time to plan and prepare nutritious meals, leading to increased consumption of healthy foods.

Having a mobile phone was significantly associated with NCD-risk scores. Mobile phones enable exposure to digital marketing and advertising, potentially influencing food choices and consumption patterns [35]. Additionally, women who own mobile phones may be more influenced by friends or family members to consume unhealthy foods. They could also be doing business outside the home, hence purchasing unhealthy foods sold around by the food vendors, which are mainly french fries, African doughnuts, and fried bananas or cassava. Findings of this study are inconsistent with those reported in a study conducted in Ethiopia and Uganda, which revealed that owning a mobile phone was significantly associated with a wide intake of a variety of food groups due to access to nutrition and health information [36,37].

## Factors associated with minimum dietary diversity for women of reproductive age

Regression analysis indicated that being older was associated with MDD-W. This can possibly be explained by the fact that older individuals often have more diverse food preferences and culinary experiences accumulated over time [38]. Economic stability and access to a wider range of foods may also increase with age, allowing older women to have more opportunities to diversify their diets. Findings of this study are in line with the studies conducted in Lesotho, Kenya and India which reported older individuals having a higher minimum dietary diversity compared to their younger counterparts [39,40]. Additionally, as people age, they may develop a greater awareness of the importance of a varied diet for overall health and well-being. It is also possible that older people may have more income from the economic activities they have established over the years than the younger ones which enable then to diversify foods more. They may also receive some support from their children who are working outside the village.

Higher-income households can contribute to greater dietary diversity due to increased access to a variety of nutritious foods. High-income households often have the financial means to purchase a wider range of fruits, vegetables, proteins, and other essential nutrients, promoting a more diverse and balanced diet [41]. Additionally, economic stability reduces food insecurity, allowing for consistent access to different food groups, which is crucial for maintaining nutrition status. The findings of this study are supported by a study in Addis Ababa, Ethiopia [42] and Lesotho [39].

Having a mobile phone in rural areas among women could be associated with higher dietary diversity due to improved access to information about nutrition, agricultural practices, and market prices. Mobile phones enable women to receive timely updates on farming techniques, access weather forecasts, and connect with markets, allowing them to make informed decisions about crop choices and food purchases [43]. The findings of this study are in line with the study conducted in Jerdu, Ethiopia by [36] which found that women who own mobile phones were more likely to have a minimum dietary diversity score than their counterparts. Moreover, mobile phones can facilitate communication within communities, creating opportunities for knowledge-sharing related to diverse and nutritious food options among women.

Minimum dietary diversity score for women was higher in male headed households than female headed ones. This can possibly be explained by male heads of households may have greater access to resources, which can facilitate the purchase of healthier food options and wide variety of foods [1]. It might as well be related to the demand of household chores which may limit the time

and energy to plan and prepare a wide variety of foods in female headed households. Findings of this study are supported by the study done in Tanzania and Burkina Faso, which found that MDD-W was lower in female headed households than their counterparts [1,44].

In addition, number of children in the households was inversely associated with minimum dietary diversity. This finding can possibly be explained by the fact that a smaller number of children in households may lead to greater dietary diversity, there being more resources available per child, allowing for a wider range of food options. Additionally, smaller families might be more flexible in adapting to diverse food preferences and accommodating a wider range of foods. This contrasts with larger families, where resource constraints and time limitations might result in a more limited variety of foods. Studies done in Nakuru county Kenya and Lesotho reported a similar finding that household size can also influence adequate dietary diversity among women of reproductive age [38,39].

## Conclusion

Generally, the mean NCD-protect score was higher than the NCD-risk score, implying that rural women are more protected from NCDs as far as food intake is concerned. Although the mean NCD-risk was low, about one out of three women consumed deep-fried foods in the day preceding the survey. Household income, mobile ownership, age, sex of the household head, and respondents' number of children were strongly associated with NCD-protect and risk scores. Only one in three women met the recommended MDD-W implying that majority may have inadequate micronutrient intake. Therefore, rural communities should be supported to diversify food production and sources of income to increase purchasing power while considering proper food choices to increase micronutrients intake and avoid the risk for NCDs. The Ministry of Agriculture and other stakeholders should support and promote production of bio fortified crops and keeping small animals. The Ministry of Education should ensure that school children are aware of the NCD-risk foods and make available healthier foods choices around the school environment

It is important to note that the study was conducted in one region (Morogoro) out of 31 regions and on one district (Mvomero) out of the eight districts of Morogoro region, Tanzania; hence the data may not be generalized to other regions due to variations across the regions.

## Acknowledgements

We acknowledge the study participants for accepting to participate in the study. We appreciate the support from the field assistants and we thank the regional, district and Sokoine University of Agriculture authorities for providing support to carry out the study. We acknowledge support from FoodLAND project members who contributed to this study in one way or the other. Many thanks to Custodio E. (Instituto de Salud Carlos III, Spain) for her support in preparing the nutritional indicators and to Marini Govigli, V., Alboni, F. (University of Bologna) for coordinating the preparation of the rural consumer research procedures and questionnaires.

## Author contributions

**Conceptualization:** Akwilina Wendelin Mwanri, Fulgence Mishili, Theresia Assenga, Rashid Suleiman, Dismas Mwaseba, Susan Nchimbi-Msolla.

**Data curation:** Akwilina Wendelin Mwanri, Fulgence Mishili, Theresia Assenga, Rashid Suleiman, Dismas Mwaseba, Susan Nchimbi-Msolla.

**Formal analysis:** Akwilina Wendelin Mwanri, Fulgence Mishili, Theresia Assenga.

**Funding acquisition:** Akwilina Wendelin Mwanri, Rashid Suleiman, Susan Nchimbi-Msolla.

**Investigation:** Akwilina Wendelin Mwanri, Fulgence Mishili, Theresia Assenga, Rashid Suleiman, Dismas Mwaseba, Susan Nchimbi-Msolla.

**Methodology:** Akwilina Wendelin Mwanri, Fulgence Mishili, Theresia Assenga, Rashid Suleiman, Dismas Mwaseba, Susan Nchimbi-Msolla.

**Project administration:** Akwilina Wendelin Mwanri, Fulgence Mishili, Dismas Mwaseba.

**Supervision:** Akwilina Wendelin Mwanri, Susan Nchimbi-Msolla.

**Validation:** Akwilina Wendelin Mwanri, Fulgence Mishili, Theresia Assenga, Rashid Suleiman, Susan Nchimbi-Msolla.

**Visualization:** Akwilina Wendelin Mwanri, Fulgence Mishili, Theresia Assenga, Rashid Suleiman, Susan Nchimbi-Msolla.

**Writing – original draft:** Akwilina Wendelin Mwanri, Theresia Assenga, Dismas Mwaseba.

**Writing – review & editing:** Akwilina Wendelin Mwanri, Fulgence Mishili, Theresia Assenga, Rashid Suleiman, Dismas Mwaseba, Susan Nchimbi-Msolla.

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
