## [Decision Letter · Decision Letter 0]

22 May 2024

PONE-D-24-02702Consumption of protective and risk food groups for non-communicable diseases among women of reproductive age in rural areas of Morogoro, TanzaniaPLOS ONE

Dear Dr. Mwanri,

Thank you for submitting your manuscript to PLOS ONE. After careful consideration, we feel that it has merit but does not fully meet PLOS ONE’s publication criteria as it currently stands. Therefore, we invite you to submit a revised version of the manuscript that addresses the points raised during the review process.

**Please review carefully the helpful suggestions made by the two reviewers, and particularly please ensure you use terminology that is both consistent throughout the paper, and consistent with international usage for the measures employed.**

We look forward to receiving your revised manuscript.

Kind regards,

Susan Horton

Academic Editor

PLOS ONE

Journal Requirements:

This research is part of the Food and Local, Agricultural and Nutritional Diversity (FoodLAND) research project that has received funding from the European Union’s Horizon 2020 research and innovation programme under grant agreement [GA No 862802]. 

Additional Editor Comments:

This is an interesting topic, and use of the MDD-W and the GDR score (NCD-protect and NCD-risk) is good - these (relatively) new measures have potential for assessing diet, and there are as yet relatively few studies using these new measures. Both reviewers have suggested ways to tighten up the analysis. It is important to use consistent terminology throughout (Reviewer 1 notes that the MDD-W isn't always spelled out accurately, and Reviewer 2 notes that NCD-protect and NCD-risk are not always spelled out correctly. The sum of NCD-risk and NCD-protect should add up to the GDR score, which national data for Tanzania suggest is about 10 (see https://www.dietquality.org/indicators/gdr-score/map) and it seems as though the score obtained here would be substantially lower.

Please respond to the careful comments of the two reviews when revising your paper.

Reviewers' comments:

Reviewer's Responses to Questions

**Comments to the Author**

1. Is the manuscript technically sound, and do the data support the conclusions?

Reviewer #1: Yes

Reviewer #2: Yes

2. Has the statistical analysis been performed appropriately and rigorously? 

Reviewer #1: No

Reviewer #2: No

3. Have the authors made all data underlying the findings in their manuscript fully available?

Reviewer #1: No

Reviewer #2: No

4. Is the manuscript presented in an intelligible fashion and written in standard English?

Reviewer #1: Yes

Reviewer #2: Yes

5. Review Comments to the Author

**Reviewer #1:**  There is a great need for context-specific evidence on diet quality of women (and men) in countries like Tanzania. This study utilizes the Diet Quality Questionnaire, a tool that has been specifically adapted for Tanzania, to assess the quality of diet among 512 women of reproductive age with a child 6-23 months in one district of Morogoro region. They assess the proportion of women who meet the criteria for minimum diet diversity as well as the scores for the two metrics used to assess NCD-risk and NCD-protective dietary intake. The authors use regression models to assess the association of these scores with a range of socioeconomic dependent variables. Their findings show relatively low dietary diversity in this sample of women, as well as relatively low scores for both NCD-risk and NCD-protective measures. The results are largely consistent with the nationally representative survey conducted in 2021 by Gallup Poll of 1000 women and men in both urban and rural areas of Tanzania, although the authors of this study do not seem to be aware of that data.

The submitted manuscript describes the study background, methods and key findings adequately. The manuscript would benefit from more detailed description of the methods used in a couple of ways - what time of year the data were collected (especially with respect to seasonal food availability), how random sampling of the women across the villages was carried out, and who conducted the data collection (number of people, training provided, any assessment of inter-rater reliability). With respect to the results, it would be helpful to have a full description of the MDD-W results by dependent variable (Table 6a) - see my suggestions in the attached PDF file. Furthermore, the results of the logistic regression model used to explore the association of the dependent variables with MDD-W would be easier to interpret if the coefficients were presented as odds ratios with 95% confidence intervals.

I noted that the minimum dietary diversity indicator is frequently incorrectly described as 'medium dietary diversity' throughout the manuscript. Also, there were several instances where the in-text citation of a study was done incorrectly. Please add the name of the study author when referring to an article in the text rather than saying "a study done by [19]" - example from line 247.

In both the results and discussion sections, I found that it would be helpful to consider these women more holistically, looking at patterns or groups within the dataset. For example, rather than looking at independent associations of age and mobile phone ownership, were there patterns across these characteristics? Were the women who owned mobile phones also more likely to be younger women? Similarly for the discussion of NCD-protect and MDD-W, there is a large amount of overlap in these two metrics of diet quality. While they are distinct indicators designed to be interpreted in different ways, there did not seem to be a recognition that the same dietary intake patterns that drive one also drive the other. Rather than discussing them separately, it would be helpful to consider what are the common underlying factors that make it difficult for women to consume more fruits and vegetables, for example.

Finally, I found the discussion of implications of these results a bit sparse, limited to one sentence in the Conclusion section. Rather than placing all the responsibility on these women and their communities for increased production, increased incomes and better food choices, what are the potential implications for agriculture, food and healthy diet policies in the country? Given that households may start to purchase more food as incomes increase, what are the options for ensuring that more diverse and healthy foods are available for purchase in this district, even if it is not possible for the communities to produce all of it themselves? What role can public health actors play in providing nutrition education to all ages, including young people, to enhance awareness of foods that are associated with increased risk of NCDs? And given the challenge in meeting nutrient adequacy with locally available foods, how can the country's food fortification agenda be scaled up to benefit women in these communities? These are some possible questions for the authors to consider as they expand a bit more on what can be done in response to these findings.

Please see the attached file with additional comments and edits.

**Reviewer #2: ** Abstract:

Line 29: prevalence of overweight or obesity: overweight and obesity are not the same as they have different BMI cut-offs, so I would suggest rephrasing it such that they are complement (i.e. AND , not OR).

Line 30-31 the sentence on limited studies on diet quality: the mention of diet quality here is quite abrupt without any earlier indication on how diet is/might be related to malnutrition/overweight.

Results: the statistics on education and owning mobile phone are not results of the study, rather they are descriptive statistics of the sample. Vegetables and fried food consumption need to be defined, for example, did 70% consumed vegetables everyday? Does that imply that the remaining 30% do not consume vegetables at all?

Line 43: NCD-risk and protect score needs a brief explanation.

Line 44-45 mentions a number of factors that are associated with NCD-protect and risk scores. Given that one is a protection score and the other a risk score, the same household factors cannot be associated with both, without mentioning the direction of the association.

Introduction:

Line 55: remove “a” before “low birth weight”.

Line 62: does it imply that undernutrition does not at all affect those who are not in the mentioned group at all?

Line 66-67: sentence unclear/check grammar.

Line 73-74 needs reference.

Line 78: by [12]- incorrect method of citation. Similarly in line 247, 278, and others.

Line 80: what is the age range for women of reproductive age?

Description of study area: this section can be made concise, e.g. the information on latitude, longitude and rainfall and irrigation schemes are somewhat unnecessary. More importantly, is the district primarily urban or rural?

Line 98: unclear/confusing sentence, consider rephrasing.

Data collection and sampling:

the FoodLAND project is introduced abruptly without description or future mention of it.

This section is 518 mothers while the abstract says 512.

Econometrics model:

Line 147: “a single model with two dependent variables…” is misleading, kindly rephrase.

I would assume that owning a mobile phone is correlated with household income, and if this is true, there is a possibility of collinearity between them. Similarly, income might be correlated to the household head being a male, explaining the results later that both income and having a male household head are associated with higher NCD-protect and risk scores and dietary diversity. I would suggest examining possible correlations between your dependent variables and if there is indeed a strong one, collinearity needs to be accounted for in the regression.

The variable names in equation 1 and 2 need to be defined.

Results: the association between the dependent variables and the sociodemographic variables needs to be unpacked and explained (direction and strength of the relationship and what they actually mean).

Line 250: check grammar.

Line 256-260: these two sentences are difficult to understand.

Line 279-282: I would avoid making a comparison between findings from rural Tanzania and that from the US and UK, as they have a completely different demography, level of development, and food market.

Some areas mention “NCD-protect” and sometimes “NCD-protective” is used.

Line 349: male household heads may also have more resources, not just access? Refer to my earlier comment on possible correlation between male household head and household income. Is there any literature that shows male household heads is associated with better nutrition in the family?

Line 369-370: see earlier comment in abstract.

6. PLOS authors have the option to publish the peer review history of their article (what does this mean? ). If published, this will include your full peer review and any attached files.

**Do you want your identity to be public for this peer review?** For information about this choice, including consent withdrawal, please see our Privacy Policy .

Reviewer #1: No

Reviewer #2: **Yes: ** Daphne C. Wu

---

## [Author Response · Author response to Decision Letter 1]

16 Oct 2024

The response to the reviewers comments is attached with the rebutal letter. We would like to highlight some responses to the comments as follows:

1. The role of the funders is stated in the submission letter. It is also included in source of funds section in the manuscript

2. The data will be made available. The project consortium is aware of the requirement and ready to share data when needed

3. Amendment of the abstract was done to make the one online identical with the manuscript

4. The ethics statement in the abstract was deleted

5. We appreciate the comment on the additional manuscript published in 2021. We included it to improve our discussion

6. The mothers were selected randomly from the village registry which had a list of all women with children of age 6 to 23 months. 10 enumerators (all females) were trained to collect data. The data collection tool was pretested and checking for consistency in asking questions. Table 6 on the MDD-W was improved to include OR and CI as suggested (Table 6b); Pg 16

7. Thank you for the observation. We admit this incorrect way of citation. It was corrected throughout the manuscript

8. It is true that the women who owned mobile phones were younger. Generally, 50% of those of age 15 to 25 had mobile phones compared to 36% among 26 to 35 years and 14% for those older than 35 years

However, all this variables were entered in the model together to control for confounding effect. The discussion section was revised as suggested

9. The conclusion was improved as suggested. It was edited in the abstract and in the final conclusion of the paper

Response to other comments is included in the rebutal letter

---

## [Editor Report · Decision Letter 1]

20 Oct 2024

PONE-D-24-02702R1Consumption of protective and risk food groups for non-communicable diseases among women of reproductive age in rural areas of Morogoro, TanzaniaPLOS ONE

Dear Dr. Mwanri,

Thank you for submitting your manuscript to PLOS ONE. After careful consideration, we feel that it has merit but does not fully meet PLOS ONE’s publication criteria as it currently stands. Therefore, we invite you to submit a revised version of the manuscript that addresses the points raised during the review process.

 We have made suggestions as to how to bring the manuscript to the required professional level of writing. In the course of revisions you may also find other examples of similar errors (e.g. plural verbs with singular nouns, or singular verbs with plural nouns; inconsistencies in the way a specific term or acronym is written) and if so please also correct ones we may have missed. 

We look forward to receiving your revised manuscript.

Kind regards,

Susan Horton

Academic Editor

PLOS ONE

Journal Requirements:

Additional Editor Comments:

Thank you for responding to the two Reviews' comments. As Associate Editor, I am suggesting a substantial number of edits to make this into a professional-quality publication. It is extremely helpful for readers to be absolutely consistent in use of abbreviations, title, etc. I have gone through in detail and suggested many such changes, but please also do your own checking in case there are other examples I have missed. Using the "edit" and "find" commands in Word and going through the whole document can be helpful.

Here are my detailed suggestions:

All line numbers refer to originally submitted manuscript with “track changes” marked (not the revised clean version or the original clean version).

Line 37: “A structure questionnaire was used to collect demographic data and the Global Diet Quality Questionnaire was used to collect dietary intake dat for women” instead of “A structured questionnaire and the Global Diet Quality Questionnaire for women were used to collect demographic and dietary intake data”

Line 47: “employment or self-employment in agriculture” not “agriculture”

Line 52: suggest you define non-communicable diseases as “NCDs” not “NCD” and then use this consistently throughout

Line 52: Capitalize Agriculture (Ministry of Agriculture)

Line 64: retain word “both”

Line 75: suggest “and of underweight is 10%” not “and underweight of 10%”

Line 84: suggest “fat, salt” not “fat salt”

Line 94: use acronym “NCDs” since you have defined it above (and for any subsequent references)

Line 128: suggest “minimum dietary diversity score for women (MDD-W)” not “minimum dietary diversity score” (since MDD usually refers to the score for children). Then do a search and replace throughout to consistently abbreviate minimum dietary diversity score to MDD-W thereafter.

Line 130 “are free” not “it is free” (tools is plural)

Line 133: suggest you are consistent in the titles of the scores: e.g. “NCD-protect” and “NCD-risk” throughout (or “NCD-Protect” and “NCD-Risk” as in lines 134 and 140) but don’t switch between them. Do a search and replace. Also, keep them in the same order throughout when talking about both scores e.g. NCD-protect and risk, don’t switch to NCD-risk and protect as occurs at subsequent points in the text.

Line 154: “were used” not “was used” (variables, used in line 153, is plural)

Line 156 “Econometric model” not “Econometrics model”

Line 161: “were analyzed” (models is plural)

Line 171: delete “monthly” from food expenditure – if the expenditure is all monthly then define it that way at first usage, and thereafter do not add “monthly”. Need to be consistent

Lines 174 and 186: suggest “Agricultural Employment/Self-employment” – if defined here, then you can use “agricultural” as a short form later

Lines 175 and 187: “Sex of head of household is Male” (if that is correct)

Line 188: suggest you use title Minimum Dietary Diversity for Women; and you don’t need to redefine it in line 189 and 193 as it has been defined above.

Line 191: suggest add “:” before commencing the list

Line 193: suggest end the list with “and other vegetables and fruit”

Line 199: I think you mean “socio-demographic” not “social demographic”

Line 219: if you have redefined 0 monthly income as missing, then you need to give the actual minimum in line 219 and in the table below. Also in line 219, replace “1,550,000” with “1,550,000 TZS” for clarity, and state “mean monthly household income” not “average household income”

Line 213: suggest “26.4, with a standard deviation of 7.3” not “26.4 +/- 7.3” – this isn’t a confidence interval.

Line 229 and Table 3: use term baked/ grain-based sweets consistently (same format as QDQQ), here and elsewhere.

Line 241: Table 4b: add in table header NCD-Protect (mean+/-SD) and NCD-Risk (mean +/- SD). Also integrate table 4a as 2 additional lines in table 4b. You don’t need to retain the data on the ranges, since you don’t have those in table 4b.

Table 5: either state “Agricultural” or “Agricultural employment/self-employment” to be consistent. Also reorganize lines 153 and 154 as 4 additional lines of Table 5 (one row for RMSE, R-square and F. You can add asterisks to indicate significance to the F statistics, and then no row for P is required.

Table 6: make title Minimum Dietary Diversity for Women Score and replace DDS by MDD-W within the table

Renumber Table 6b as Table 7 (and change text references consistently). Consider carefully whether you really need 4 decimal places throughout table. 2 will likely suffice. Also relabel “Agicultural employment” either as “Agricultural” or “Agricultural employment/self-employment”

Line 321: “younger counterparts” not “counterparts”

Line 345: “NCD-risk” not “NCDs-risk”

Line 350: “french fries” not “French fries”

Line 355: “minimum” not “medium”

Line 387: “chores which” not “chores”

Line 390: “MDD-W” not “minimum dietary diversity”; note that this change should be made throughout. MDD refers to children’s diet diversity, and data on this has not been presented in this study. In line 398, do these other studies look at “adequate dietary diversity” or do they look at MDD-W?

Line 407: “MDD-W” not “MDD) – same as previous comment.

Line 411: Ministry of Agriculture” not “Ministry of agriculture”; similar comment re “ministry of education” in following sentence.

---

## [Author Response · Author response to Decision Letter 2]

18 Dec 2024

We appreciate comments from the Associate Editor. We have considered the comments and suggestions as guided. We hereby re-submit for consideration. Looking forward to your positive response.

---

## [Decision Letter · Decision Letter 2]

21 Jan 2025

Consumption of protective and risk food groups for non-communicable diseases among women of reproductive age in rural areas of Morogoro, Tanzania

PONE-D-24-02702R2

Dear Dr. Mwanri,

We’re pleased to inform you that your manuscript has been judged scientifically suitable for publication and will be formally accepted for publication once it meets all outstanding technical requirements.

Kind regards,

Susan Horton

Academic Editor

PLOS ONE

Additional Editor Comments (optional):

You may wish to provide the exact reference for calculating NCD-risk and NCD-protect which is https://drive.google.com/file/d/1eplRm9i5_109-a5Ac1Lqj-lUI3VgVIFx/view It can probably be found somehow from reference [15] in the paper, but would take effort.

Please also consider the following small edits. Line numbers below refer to tracked changes version of revised document. Reviewer 1 has also suggested some small edits.

Line 37 (abstract): “structured” not “structure”

Line 47: “NCD-protect” not “protect”

Line 49 “with NCD-protect” not “NCD protect”

Line 196: “Ten food groups that include” not “includes”

Bottom of table 5: the coefficient on the constant is also significant (add asterisks). Same in table 7.

Lines 288-289. What does “consumption in by 97%” mean? Are there missing words here?

Reviewers' comments:

Reviewer's Responses to Questions

**Comments to the Author**

1. If the authors have adequately addressed your comments raised in a previous round of review and you feel that this manuscript is now acceptable for publication, you may indicate that here to bypass the “Comments to the Author” section, enter your conflict of interest statement in the “Confidential to Editor” section, and submit your "Accept" recommendation.

Reviewer #1: (No Response)

2. Is the manuscript technically sound, and do the data support the conclusions?

Reviewer #1: Yes

3. Has the statistical analysis been performed appropriately and rigorously? 

Reviewer #1: Yes

4. Have the authors made all data underlying the findings in their manuscript fully available?

Reviewer #1: (No Response)

5. Is the manuscript presented in an intelligible fashion and written in standard English?

Reviewer #1: Yes

6. Review Comments to the Author

Reviewer #1: The majority of concerns have been addressed by the authors. There are still a few minor edits to enhance the quality of the final version, which I have suggested in the attached file.

7. PLOS authors have the option to publish the peer review history of their article (what does this mean? ). If published, this will include your full peer review and any attached files.

**Do you want your identity to be public for this peer review?** For information about this choice, including consent withdrawal, please see our Privacy Policy .

Reviewer #1: No

---

## [Editor Report · Acceptance letter]

PONE-D-24-02702R2

PLOS ONE

Dear Dr. Mwanri,

I'm pleased to inform you that your manuscript has been deemed suitable for publication in PLOS ONE. Congratulations! Your manuscript is now being handed over to our production team.

Kind regards,

on behalf of

Dr. Susan Horton

Academic Editor

PLOS ONE